# AUTOMATED GENERATION OF MULTILINGUAL CLUSTERS FOR THE EVALUATION OF DISTRIBUTED REPRESENTATIONS

**Philip Blair, Yuval Merhav & Joel Barry**
Basis Technology
One Alewife Center
Cambridge, MA 02140 USA
{pblair,yuval,joelb}@basistech.com

## ABSTRACT

We propose a language-agnostic way of automatically generating sets of semantically similar clusters of entities along with sets of "outlier" elements, which may then be used to perform an intrinsic evaluation of word embeddings in the *outlier detection* task. We used our methodology to create a gold-standard dataset, which we call WikiSem500, and evaluated multiple state-of-the-art embeddings. The results show a correlation between performance on this dataset and performance on sentiment analysis.

## 1 INTRODUCTION

High quality datasets for evaluating word and phrase representations are essential for building better models that can advance natural language understanding. Various researchers have developed and shared datasets for syntactic and semantic intrinsic evaluation. The majority of these datasets are based on word similarity (e.g., Finkelstein et al. (2001); Bruni et al. (2012); Hill et al. (2016)) and analogy tasks (e.g., Mikolov et al. (2013a;b)). While there has been a significant amount of work in this area which has resulted in a large number of publicly available datasets, many researchers have recently identified problems with existing datasets and called for further research on better evaluation methods (Faruqui et al., 2016; Gladkova et al., 2016; Hill et al., 2016; Avraham & Goldberg, 2016; Linzen, 2016; Batchkarov et al., 2016). A significant problem with word similarity tasks is that human bias and subjectivity result in low inter-annotator agreement and, consequently, human performance that is lower than automatic methods (Hill et al., 2016). Another issue is low or no correlation between intrinsic and extrinsic evaluation metrics (Chiu et al., 2016; Schnabel et al., 2015).

Recently, Camacho-Collados & Navigli (2016) proposed the *outlier detection task* as an intrinsic evaluation method that improved upon some of the shortcomings of word similarity tasks. The task builds upon the "word intrusion" task initially described in Chang et al. (2009): given a set of words, the goal is to identify the word that does not belong in the set. However, like the vast majority of existing datasets, this dataset requires manual annotations that suffer from human subjectivity and bias, and it is not multilingual.

Inspired by Camacho-Collados & Navigli (2016), we have created a new outlier detection dataset that can be used for intrinsic evaluation of semantic models. The main advantage of our approach is that it is fully automated using Wikidata and Wikipedia, and it is also diverse in the number of included topics, words and phrases, and languages. At a high-level, our approach is simple: we view Wikidata as a graph, where nodes are entities (e.g., ⟨Chicago Bulls, Q128109⟩, ⟨basketball team, Q13393265⟩), edges represent "instance of" and "subclass of" relations (e.g., ⟨Chicago Bulls, Q128109⟩ is an instance of ⟨basketball team, Q13393265⟩, ⟨basketball team, Q13393265⟩ is a subclass of ⟨sports team, Q12973014⟩), and the semantic similarity between two entities is inversely proportional to their graph distance (e.g., ⟨Chicago Bulls, Q128109⟩ and ⟨Los Angeles Lakers, Q121783⟩ are semantically similar since they are both instance of ⟨basketball team, Q13393265⟩). This way we can form semantic clusters

by picking entities that are members of the same class, and picking outliers with different notions of dissimilarity based on their distance from the cluster entities.

We release the first version of our dataset, which we call WikiSem500, to the research community. It contains around 500 per-language cluster groups for English, Spanish, German, Chinese, and Japanese (a total of 13,314 test cases). While we have not studied yet the correlation between performance on this dataset and various downstream tasks, our results show correlation with sentiment analysis. We hope that this diverse and multilingual dataset will help researchers to advance the state-of-the-art of word and phrase representations.

## 2  RELATED WORK

Word similarity tasks have been popular for evaluating distributional similarity models. The basic idea is having annotators assigning similarity scores for word pairs. Models that can automatically assign similarity scores to the same word pairs are evaluated by computing the correlation between their and the human assigned scores. Schnabel et al. (2015) and Hill et al. (2016) review many of these datasets. Hill et al. (2016) also argue that the predominant gold standards for semantic evaluation in NLP do not measure the ability of models to reflect similarity. Their main argument is that many such benchmarks measure association and relatedness and not necessarily similarity, which limits their suitability for a wide range of applications. One of their motivating examples is the word pair "coffee" and "cup," which have high similarity ratings in some benchmarks despite not being very similar. Consequently, they developed guidelines that distinguish between association and similarity and used five hundred Amazon Mechanical Turk annotators to create a new dataset called SimLex-999, which has higher inter annotator agreement than previous datasets. Avraham & Goldberg (2016) improved this line of work further by redesigning the annotation task from rating scales to ranking, in order to alleviate bias, and also redefined the evaluation measure to penalize models more for making wrong predictions on reliable rankings than on unreliable ones.

Another popular task based on is word analogies. The analogy dataset proposed by Mikolov et al. (2013a) has become a standard evaluation set. The dataset contains fourteen categories, but only about half of them are for semantic evaluation (e.g. "US Cities", "Common Capitals", "All Capitals"). In contrast, WikiSem500 contains hundreds of categories, making it a far more diverse and challenging dataset for the general-purpose evaluation of word representations. The Mikolov dataset has the advantage of additionally including *syntactic* categories, which we have left for future work.

Camacho-Collados & Navigli (2016) addressed some of the issues mentioned previously by proposing the outlier detection task. Given a set of words, the goal is to identify the word that does not belong in the set. Their pilot dataset consists of eight different topics each made up of a cluster of eight words and eight possible outliers. Four annotators were used for the creation of the dataset. The main advantage of this dataset is its near perfect human performance. However, we believe a major reason for that is the specific choice of clusters and the small size of the dataset.

## 3  GENERATING THE DATASET

In a similar format to the one used in the dataset furnished by Camacho-Collados & Navigli (2016), we generated sets of entities which were semantically similar to one another, known as a "cluster", followed by up to three pairs (as available) of dissimilar entities, or "outliers", each with different levels of semantic similarity to the cluster. The core thesis behind our design is that our knowledge base, Wikidata (2016), can be treated like a graph, where the semantic similarity between two elements is inversely proportional to their graph distance.

Informally, we treat Wikidata entities which are instances of a common entity as a cluster (see Figure 1). Then, starting from that common entity (which we call a 'class'), we follow "subclass of" relationships to find a sibling class (see "American Football Team" in Figure 1). Two items which are instances of the sibling class (but *not* instances of the original class) are chosen as outliers. The process is then repeated with a 'cousin' class with a common *grandparent* to the original class (see "Ice Hockey Team" in Figure 1). Finally, we choose two additional outliers by randomly selecting items which are a distance of at least 7 steps away from the original class. These three "outlier classes" are referred to as $O_1$, $O_2$, and $O_3$ outlier classes, respectively.

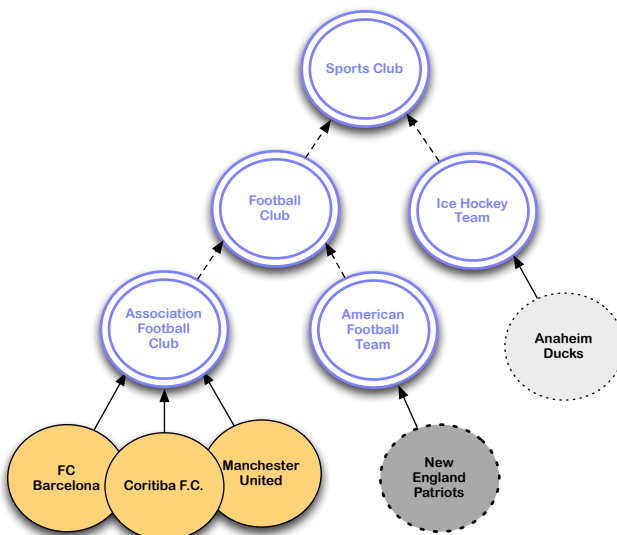

Figure 1: Partial example of a Wikidata cluster. Solid arrows represent "Instance Of" relationships, and dashed arrows represent "Subclass Of" relationships.

A full formalization of our approach is described in Appendix A.

## 3.1 REFINING THE DATASET QUALITY

Prior to developing a framework to improve the quality of the generated dataset, we performed a small amount of manual pruning of our Wikidata graph. Disambiguation pages led to bizarre clusters of entities, for their associated relationships are not true semantic connections, but are instead artifacts of the structure of our knowledge base. As such, they were removed. Additionally, classes within a distance of three from the entity for "Entity" itself[1] (Q35120) had instances which had quite weak semantic similarity (one example being "human"). We decided that entities at this depth range ought to be removed from the Wikidata graph as well.

Once our Wikidata dump was pruned, we employed a few extra steps at generation time to further improve the quality of the dataset; first and foremost were how we chose representative instances and outliers for each class (see $\sigma_i$ and $\sigma_o$ in Appendix A). While "San Antonio Spurs" and "Chicago Bulls" may both be instances of "basketball team", so are "BC Andorra" and "Olimpia Milano." We wanted the cluster entities to be as strongly related as possible, so we sought a class-agnostic heuristic to accomplish this. Ultimately, we found that favoring entities whose associated Wikipedia pages had higher sitelink counts gave us the desired effect.

As such, we created clusters by choosing the top eight instances of a given class, ranked by sitelink count. Additionally, we only chose items as outliers when they had at least ten sitelinks so as to remove those which were 'overly obscure,' for the ability of word embeddings to identify rare words (Schnabel et al., 2015) would artificially decrease the difficulty of such outliers.

We then noticed that many cluster entities had similarities in their labels that could be removed if a different label was chosen. For example, 80% of the entities chosen for "association football club" ended with the phrase "F.C." This essentially invalidates the cluster, for the high degree of syntactic overlap artificially increases the cosine similarity of all cluster items in word-level embeddings. In order to increase the quality of the surface forms chosen for each entity, we modified our resolution of entity QIDs to surface forms (see $\tau$ in Appendix A) to incorporate a variant[2] of the work from

---

[1] Q35120 is effectively the "root" node of the Wikidata graph; 95.5% of nodes have "subclass of" chains which terminate at this node.

[2] By 'variant,' we are referring to the fact that the dictionaries in which we perform the probability lookups are constructed for *each language*, as opposed to the cross-lingual dictionaries originally described by Spitkovsky & Chang (2012).

Spitkovsky & Chang (2012):

$$\tau(QID) = \arg\max_s \{P(s \mid \text{wikipedia page}(QID))\} \tag{1}$$

That is, the string for an entity is the string which is most likely to link to the Wikipedia page associated with that entity. For example, half of the inlinks to the page for Manchester United FC are the string "Manchester United," which is the colloquial way of referring to the team.

Next, we filter out remaining clusters using a small set of heuristics. The following clusters are rejected:

- Clusters with more than two items are identical after having all digits removed. This handles cases such as entities only differing by years (e.g. "January 2010," "January 2012," etc.).

- Clusters with more than three elements have identical first or last six characters[3]. Characters are compared instead of words in order to better support inflected languages. This was inspired by clusters for classes such as "counties of Texas" (Q11774097), where even the dictionary-resolved aliases have high degrees of syntactic overlap (namely, over half of the cluster items ended with the word "County").

- Clusters in which any item has an occurrence of a 'stop affix,' such as the prefix "Category:" or the suffix "一覧" (a Japanese Wikipedia equivalent of "List of"). In truth, this could be done during preprocessing, but doing it at cluster generation time instead has no bearing on the final results. These were originally all included under an additional stop class ("Wikimedia page outside the main knowledge tree") at prune time, but miscategorizations in the Wikidata hierarchy prevented us from doing so; for example, a now-removed link resulted in every country being pruned from the dataset. As such, we opted to take a more conservative approach and perform this on at cluster-generation time and fine tune our stoplist as needed.

- Clusters with more than one entity with a string length of one. This prevents clusters such as "letters of the alphabet" being created. Note that this heuristic was disabled for the creation of Chinese and Japanese clusters.

- Clusters with too few entities, after duplicates introduced by resolving entities to surface forms ($\tau$) are removed.

## 3.2 THE WIKISEM500 DATASET

Using the above heuristics and preprocessing, we have generated a dataset, which we call **WikiSem500**[4]. Our dataset is formatted as a series of files containing *test groups*, comprised of a cluster and a series of outliers. *Test cases* can be constructed by taking each outlier in a given group with that group's cluster. Table 1 shows the number of included test groups and test cases for each language. Each group contains a cluster of 7-8 entities and up to two entities from each of the three outlier classes. Table 2 shows example clusters taken from the dataset.

## 4 EVALUATION

For clarity, we first restate the definitions of the scoring metrics defined by Camacho-Collados & Navigli (2016) in terms of test groups (in contrast to the original definition, which is defined in terms of test *cases*). The way in which out-of-vocabulary entities are handled and scores are reported makes this distinction important, as will be seen in Section 4.3.

---

[3] For Chinese and Japanese, this is modified such that the at least six entities must have identical (non-kana) first or last characters, or more than three must have identical the same first or last *two* characters. Because English is not inflected, we simply use spaces as approximate word boundaries and check that the first or last of *those* does not occur too often.

[4] The dataset is available for download at https://github.com/belph/wiki-sem-500

Table 1: Statistics of the WikiSem500 dataset.

| Language | Test Groups | Test Cases |
|----------|-------------|------------|
| English | 500 | 2,816 |
| Spanish | 500 | 2,777 |
| German | 500 | 2,780 |
| Japanese | 448 | 2,492 |
| Chinese | 441 | 2,449 |

Table 2: Example partial clusters from the WikiSem500 dataset. Classes, clusters, and outliers are shown.

| | fictional country | mobile operating system | video game publisher | emotion |
|---|---|---|---|---|
| Cluster Items | Mordor | Windows Phone | Activision | fear |
| | Rohan | Firefox OS | Nintendo | love |
| | Shire | iOS | Valve Corporation | happiness |
| | Arnor | Android | Electronic Arts | anger |
| Outliers | Thule | Periscope | HarperCollins | magnitude |
| | Duat | Ingress | Random House | Gini coefficient |
| | Donkey Kong | iWeb | Death Row Records | redistribution of wealth |
| | Scrooge McDuck | iPhoto | Sun Records | Summa Theologica |

The core measure during evaluation is known as the *compactness score*; given a set $W$ of words, it is defined as follows:

$$\forall w \in W, c(w) = \frac{1}{(|W| - 1)(|W| - 2)} \sum_{w_i \in W \backslash \{w\}} \sum_{\substack{w_j \in W \backslash \{w\} \\ w_j \neq w_i}} sim(w_i, w_j) \tag{2}$$

where $sim$ is a vector similarity measure (typically cosine similarity). Note that Camacho-Collados & Navigli (2016) reduces the asymptotic complexity of $c(w)$ from $O(n^3)$ to $O(n^2)$. We denote $P(W, w)$ to be the (zero-indexed) position of $w$ in the list of elements of $W$, sorted by compactness score in descending order. From this, we can describe the following definition for *Outlier Position* (OP), where $\langle C, O \rangle$ is a test group and $o \in O$:

$$OP(C \cup \{o\}) = P(C \cup \{o\}, o) \tag{3}$$

This gives rise to the boolean-valued *Outlier Detection* (OD) function:

$$OD(C \cup \{o\}) = \begin{cases} 1 & OP(C \cup \{o\}) = |C| \\ 0 & \text{otherwise} \end{cases} \tag{4}$$

Finally, we can now describe the *Outlier Position Percentage* (OPP) and *Accuracy* scores:

$$OPP(D) = \frac{\sum_{\langle C,O \rangle \in D} \sum_{o \in O} \frac{OP(C \cup \{o\})}{|C|}}{\sum_{\langle C,O \rangle \in D} |O|} \tag{5}$$

$$Accuracy(D) = \frac{\sum_{\langle C,O \rangle \in D} \sum_{o \in O} OD(C \cup \{o\})}{\sum_{\langle C,O \rangle \in D} |O|} \tag{6}$$

### 4.1 HANDLING OUT-OF-VOCABULARY WORDS

One thing Camacho-Collados & Navigli (2016) does not address is how out-of-vocabulary (OOV) items should be handled. Because our dataset is much larger and contains a wider variety of words, we have extended their work to include additional scoring provisions which better encapsulate the performance of vector sets trained on different corpora.

There are two approaches to handling out-of-vocabulary entities: use a sentinel vector to represent all such entities or discard such entities entirely. The first approach is simpler, but it has a number of drawbacks; for one, a poor choice of sentinel can have a drastic impact on results. For example, an implementation which uses the zero vector as a sentinel and defines $sim(\vec{x}, \vec{0}) = 0 \, \forall \vec{x}$ places many non-out-of-vocabulary outliers at a large disadvantage in a number of vector spaces, for we have found that negative compactness scores are rare. The second approach avoids deliberately introducing invalid data into the testing evaluation, but comparing scores across vector embeddings with different vocabularies is difficult due to them having different in-vocabulary subsets of the test set.

We have opted for the latter approach, computing the results on both the entire dataset and on only the intersection of in-vocabulary entities between all evaluated vector embeddings. This allows us to compare embedding performance both when faced with the same unknown data and when evaluated on the same, in-vocabulary data.

### 4.2 HUMAN BASELINE

In order to gauge how well embeddings should perform on our dataset, we conducted a human evaluation. We asked participants to select the outlier from a given test case, providing us with a human baseline for the accuracy score on the dataset. We computed the non-out-of-vocabulary intersection of the embeddings shown in Table 4, from which 60 test groups were sampled. Due to the wide array of domain knowledge needed to perform well on the dataset, participants were allowed to refer to Wikipedia (but explicitly told *not* to use Wikidata). We collected 447 responses, with an overall precision of 68.9%.

The performance found is not as high as on the baseline described in Camacho-Collados & Navigli (2016), so we conducted a second human evaluation on a smaller hand-picked set of clusters in order to determine whether a lack of domain knowledge or a systemic issue with our method was to blame. We had 6 annotators fully annotate 15 clusters generated with our system. Each cluster had one outlier, with a third of the clusters having each of the three outlier classes. Human performance was at 93%, with each annotator missing exactly one cluster. Five out of the six annotators missed the same cluster, which was based on books and contained an $O_1$ outlier (the most difficult class). We interviewed the annotators, and three of them cited a lack of clarity on Wikipedia over whether or not the presented outlier was a book (leading them to guess), while the other two cited a conflation with one of the book titles and a recently popular Broadway production.

With the exception of this cluster, the performance was near-perfect, with one annotator missing one cluster. Consequently, we believe that the lower human performance on our dataset is primarily a result of the dataset's broad domain.

### 4.3 EMBEDDING RESULTS

We evaluated our dataset on a number of publicly available vector embeddings: the Google News-trained CBOW model released by Mikolov et al. (2013a), the 840-billion token Common Crawl corpus-trained GloVe model released by Pennington et al. (2014), and the English, Spanish, German, Japanese, and Chinese MultiCCA vectors[5] from Ammar et al. (2016), which are trained on a combination of the Europarl (Koehn, 2005) and Leipzig (Quasthoff et al., 2006) corpora. In ad-

---

[5]The vectors are word2vec CBOW vectors, and the non-English vectors are aligned to the English vector space. Reproducing the original (unaligned) non-English vectors yields near-identical results to the aligned vectors.

Table 3: Performance of English word embeddings on the entire WikiSem500 dataset.

| Model | Corpus | OPP | Acc. | Groups Skipped | % Cluster Items OOV | % Outliers OOV |
|---|---|---|---|---|---|---|
| GloVe | Common Crawl | 75.53 | 38.57 | 5 | 6.33 | 5.70 |
| | Wikipedia+Gigaword | 79.86 | 50.61 | 2 | 4.25 | 4.02 |
| CBOW | Wikipedia+Gigaword | 84.97 | 55.80 | 2 | 4.25 | 4.02 |
| | Google News (phrases) | 63.10 | 22.60 | 6 | 13.68 | 15.02 |
| | Google News | 65.13 | 24.45 | 6 | 13.68 | 15.02 |
| | Leipzig+Europarl | 74.59 | 42.62 | 18 | 22.03 | 19.62 |
| Skip-Gram | Wikipedia+Gigaword | 84.44 | 57.66 | 2 | 4.25 | 4.02 |

dition, we trained GloVe, CBOW, and Skip-Gram (Mikolov et al., 2013a) models on an identical corpus comprised of an English Wikipedia dump and Gigaword corpus[6].

The bulk of the embeddings we evaluated were word embeddings (as opposed to phrase embeddings), so we needed to combine each embeddings' vectors in order to represent multi-word entities. If the embedding *does* handle phrases (only Google News), we perform a greedy lookup for the longest matching subphrase in the embedding, averaging the subphrase vectors; otherwise, we take a simple average of the vectors for each token in the phrase. If a token is out-of-vocabulary, it is ignored. If all tokens are out-of-vocabulary, the entity is discarded. This check happens as a preprocessing step in order to guarantee that a test case does not have its outlier thrown away. As such, we report the percentage of cluster entities filtered out for being out-of-vocabulary *separately* from the outliers which are filtered out, for the latter results in an entire test case being discarded.

In order to compare how well each vector embedding would do when run on unknown input data, we first collected the scores of each embedding on the entire dataset. Table 3 shows the Outlier Position Percentage (OPP) and accuracy scores of each embedding, along with the number of test groups which were skipped entirely[7] and the mean percentage of out-of-vocabulary cluster entities and outliers among all test groups[8]. As in Camacho-Collados & Navigli (2016), we used cosine similarity for the $sim$ measure in Equation 2.

The MultiCCA (Leipzig+Europarl) CBOW vectors have the highest rate of out-of-vocabulary entities, likely due in large part to the fact that its vocabulary is an order of magnitude smaller than the other embeddings (176,691, while the other embeddings had vocabulary sizes of over 1,000,000). Perhaps most surprising is the below-average performance of the Google News vectors. While attempting to understand this phenomenon, we noticed that disabling the phrase vectors *boosted* performance; as such, we have reported the performance of the vectors with and without phrase vectors enabled.

Inspecting the vocabulary of the Google News vectors, we have inferred that the vocabulary has undergone some form of normalization; performing the normalizations which we can be reasonably certain were done before evaluating has a negligible impact ($\approx +0.01\%$) on the overall score. The Google News scores shown in Table 3 are *with* the normalization enabled. Ultimately, we hypothesize that the discrepancy in Google News scores comes down to the training corpus. We observe a bias in performance on our training set towards Wikipedia-trained vectors (discussed below; see Table 5), and, additionally, we expect that the Google News corpus did not have the wide regional

---

[6]We used the July 2016 Wikipedia dump (Wikimedia, 2016) and the 2011 Gigaword corpus (Parker et al., 2011).

[7]This happens when either all outliers are out-of-vocabulary or fewer than two cluster items are in-vocabulary. No meaningful evaluation can be performed on the remaining data, so the group is skipped.

[8]This includes the out-of-vocabulary rates of the skipped groups.

Table 4: Performance of English word embeddings on their common in-vocabulary intersection of the WikiSem500 dataset.

| Model | Corpus | OPP | Acc. |
|---|---|---|---|
| GloVe | Common Crawl | 76.73 | 43.25 |
| | Wikipedia+Gigaword | 76.19 | 47.69 |
| CBOW | Wikipedia+Gigaword | 82.59 | 55.90 |
| | Google News (with phrases) | 63.67 | 24.74 |
| | Google News | 66.20 | 27.43 |
| | Leipzig+Europarl (MultiCCA) | 75.01 | 42.82 |
| Skip-Gram | Wikipedia+Gigaword | 82.03 | 56.80 |

Table 5: Performance comparison of GloVe vectors trained on different corpora when evaluated on their common in-vocabulary intersection.

| Corpus | OPP | Acc. |
|---|---|---|
| Wikipedia+Gigaword | 80.03 | 54.43 |
| Wikipedia | 77.39 | 49.95 |
| Gigaword | 76.36 | 45.07 |

coverage that Wikidata has, limiting the training exposure to many of the more niche classes in the training set.

In order to get a better comparison between the embeddings under identical conditions, we then took the intersection of in-vocabulary entities across *all* embeddings and reevaluated on this subset. 23.88% of cluster entities and 22.37% of outliers were out-of-vocabulary across all vectors, with 23 test groups removed from evaluation. Table 4 shows the results of this evaluation.

The scores appear to scale roughly linearly when compared to Table 3, but these results serve as a more reliable 'apples to apples' comparison of the algorithms and training corpora.

Because Wikidata was the source of the dataset, we analyzed how using Wikipedia as a training corpus influenced the evaluation results. We trained three GloVe models with smaller vocabularies: one trained on only Gigaword, one trained on only Wikipedia, and one trained on both. The results of evaluating on the embeddings' common intersection are shown in Table 5. We observe a slight ($\approx 3.15\%$ relative change) bias in OPP scores with Wikipedia over Gigaword, while finding a significantly larger ($\approx 19.12\%$ relative change) bias in accuracy scores.

We believe that this bias is acceptable, for OPP scores (which we believe to be more informative) are not as sensitive to the bias and the numerous other factors involved in embedding generation (model, window size, etc.) can still be compared by controlling for the training corpora.

Additionally, we wanted to verify that the $O_1$ outlier class (most similar) was the most difficult to distinguish from the cluster entities, followed by the $O_2$ and $O_3$ classes. We generated three separate datasets, each with only one class of outliers, and evaluated each embedding on each dataset. Figure 2 illustrates a strong positive correlation between outlier class and both OPP scores and accuracy.

Finally, we used the non-English MultiCCA vectors (Ammar et al., 2016) to evaluate the multilingual aspect of our dataset. We expect to see Spanish and German perform similarly to the English Europarl+Leipzig vectors, for the monolingual training corpora used to generate them consisted of

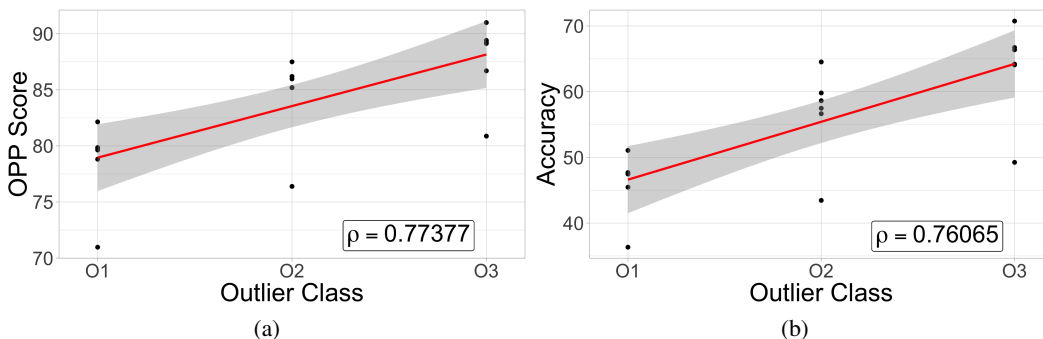

(a)　　　　　　　　　　　　　　(b)

Figure 2: OPP and accuracy scores of embeddings in Table 3 by outlier class. The Spearman $\rho$ correlation coefficients are shown.

Table 6: Performance of Non-English word embeddings on entire WikiSem500 dataset.

| Language | OPP | Acc. | Groups Skipped | % Cluster Items OOV | % Outliers OOV | Vocab. Size |
|---|---|---|---|---|---|---|
| Spanish | 77.25 | 46.00 | 22 | 21.55 | 17.75 | 225,950 |
| German | 76.17 | 43.46 | 31 | 24.45 | 25.74 | 376,552 |
| Japanese | 72.51 | 40.18 | 54 | 36.87 | 24.66 | 70,551 |
| Chinese | 67.61 | 34.58 | 12 | 37.74 | 34.29 | 70,865 |

Spanish and German equivalents of the English training corpus. Table 6 shows the results of the non-English evaluations.

We observe a high degree of consistency with the results of the English vectors. The Japanese and Chinese scores are somewhat lower, but this is likely due to their having smaller training corpora and more limited vocabularies than their counterparts in other languages.

### 4.4 CORRELATION WITH DOWNSTREAM PERFORMANCE

In light of recent concerns raised about the correlation between intrinsic word embedding evaluations and performance in downstream tasks, we sought to investigate the correlation between WikiSem500 performance and extrinsic evaluations. We used the embeddings from Schnabel et al. (2015) and ran the outlier detection task on them with our dataset.

As a baseline measurement of how well our dataset correlates with performance on alternative intrinsic tasks, we our evaluation with the scores reported in Schnabel et al. (2015) on the well-known analogy task (Mikolov et al., 2013a). Figure 3a illustrates strong correlations between analogy task performance and our evaluation's OPP scores and accuracy.

Figure 3b displays the Pearson's correlation between the performance of each embedding on the WikiSem500 dataset and the extrinsic scores of each embedding on noun-phrase chunking and sentiment analysis reported in Schnabel et al. (2015).

Similar to the results seen in the paper, performance on our dataset correlates strongly with performance on a semantic-based task (sentiment analysis), with Pearson's correlation coefficients higher than 0.97 for both accuracy and OPP scores. On the other hand, we observe a weak-to-nonexistent correlation with chunking. This is expected, however, for the dataset we have constructed consists of items which differ in semantic meaning; syntactic meaning is not captured by the dataset. It is worth noting the inconsistency between this and the intrinsic results in Figure 3a, which indicate a *stronger* correlation with the syntactic subset of the analogy task than its semantic subset. This is

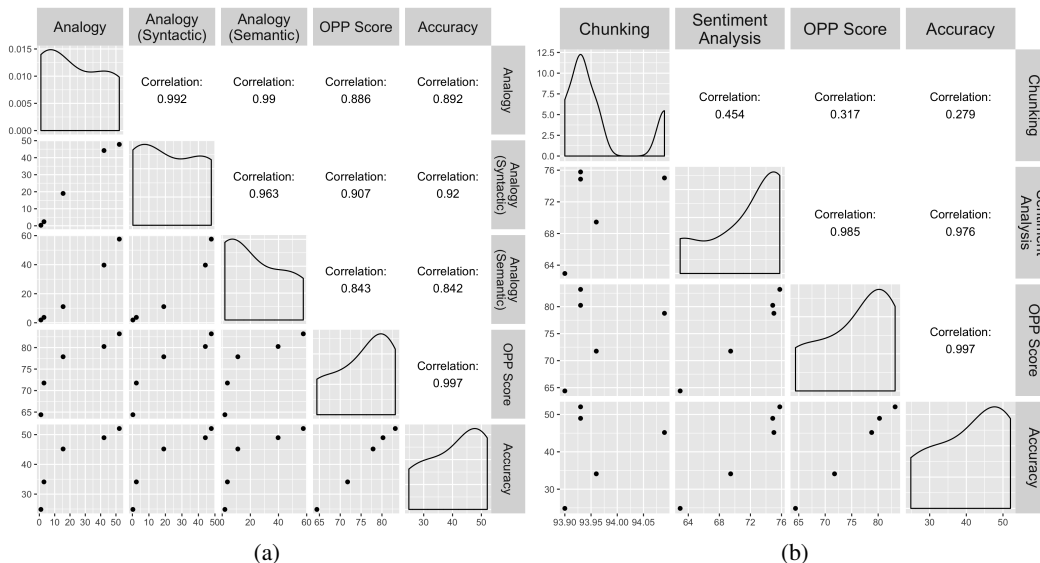

(a)                                    (b)

Figure 3: Pearson's correlation between WikiSem500 outlier detection performance and performance on the analogy task and extrinsic tasks. Distributions of values are shown on the diagonal.

expected, for it agrees with the poor correlation between chunking and intrinsic performance shown in Schnabel et al. (2015).

## 5 FUTURE WORK

Due to the favorable results we have seen from the WikiSem500 dataset, we intend to release test groups in additional languages using the method described in this paper. Additionally, we plan to study further the downstream correlation of performance on our dataset with additional downstream tasks.

Moreover, while we find a substantial correlation between performance on our dataset and on a semantically-based extrinsic task, the relationship between performance and syntactically-based tasks leaves much to be desired. We believe that the approach taken in this paper to construct our dataset could be retrofitted to a system such as WordNet (2010) or Wiktionary (2016) (for multilingual data) in order to construct *syntactically* similar clusters of items in a similar manner. We hypothesize that performance on such a dataset would correlate much more strongly with syntactically-based extrinsic evaluations such as chunking and part of speech tagging.

## 6 CONCLUSION

We have described a language-agnostic technique for generating a dataset consisting of semantically related items by treating a knowledge base as a graph. In addition, we have used this approach to construct the **WikiSem500** dataset, which we have released. We show that performance on this dataset correlates strongly with downstream performance on sentiment analysis. This method allows for creation of much larger scale datasets in a larger variety of languages without the time-intensive task of human creation. Moreover, the parallel between Wikidata's graph structure and the annotation guidelines from Camacho-Collados & Navigli (2016) preserve the simple-to-understand structure of the original dataset.

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

## A FORMALIZATION

We now provide a formal description of the approach taken to generate our dataset.

Let $V$ be the set of entities in Wikidata. For all $v_1, v_2 \in V$, we denote the relations $v_1 \prec_I v_2$ when $v_1$ is an instance of $v_2$, and $v_1 \prec_S v_2$ when $v_1$ is a subclass of $v_2$. We then define $I : V \to V^*$ as the following 'instances' mapping:

$$I(v) = \{v' \in V \mid v' \prec_I v\} \tag{7}$$

For convenience, we then denote $C = \{v \in V \mid |I(v)| \geq 2\}$; the interpretation being that $C$ is the set of entities which have enough instances to possibly be viable clusters. We now formally state the following definition:

**Definition 1.** *A set $A \subseteq V$ is a **cluster** if $A = I(v)$ for some $v \in C$. We additionally say that $v$ is the **class** associated with the cluster $A$.*

Let $P : V \to V^*$ be the following 'parent of' mapping:

$$P(v) = \{v' \in V \mid v \prec_S v'\} \tag{8}$$

Furthermore, let $P^{-1} : V \to V^*$ be the dual of $P$:

$$P^{-1}(v) = \{v' \in V \mid v' \prec_S v\} \tag{9}$$

For additional convenience, we denote the following:

$$P^k(v) = \begin{cases} P(v) & k = 1 \\ \bigcup_{v' \in P(v)} P^{k-1}(v') & k > 1 \end{cases} \tag{10}$$

As an abuse of notation, we define the following:

$$I^*(v) = I(v) \cup \left( \bigcup_{v' \in P^{-1}(v)} I^*(v) \right) \tag{11}$$

That is, $I^*(v)$ is the set of all instances of $v$ *and* all instances of anything that is a subclass of $v$ (recursively).

We then define the measure $d : V \times V \to \mathbb{N}$ to be the graph distance between any entities in $V$, using the following set of edges:

$$E_{SU} = \{(v_1, v_2) \mid v_1 \prec_S v_2 \vee v_2 \prec_S v_1\} \tag{12}$$

Finally, we define[9] three additional mappings for outliers parametrized[10] by $\mu \in \mathbb{N}^+$:

$$O_1(v) = \left( \bigcup_{p \in P(v)} \left( \bigcup_{c \in P^{-1}(p) \setminus \{v\}} I^*(c) \right) \right) \setminus I(v) \tag{13}$$

---

[9]For the definition of $O_2$, note that we do *not* say that it must be true that $p \in P^2(v) \setminus P(v)$. In practice, however, avoiding (if not excluding) certain values of $p$ in this manner can help improve the quality of resulting clusters, at the cost of reducing the number of clusters which can be produced.

[10]The WikiSem500 dataset was generated with a value of $\mu = 7$.

$$O_2(v) = \left( \bigcup_{p \in P^2(v)} \left( \bigcup_{c \in P^{-1}(p) \setminus \{v\}} I^*(c) \right) \right) \setminus I(v) \tag{14}$$

$$O_3(v) = \left( \bigcup_{p \in P(v)} \{e \in I(v') \mid \mu \le d(p, v')\} \right) \setminus I(v) \tag{15}$$

To simplify the model, we assume that all three of the above sets are mutually exclusive. Given these, we can formally state the following definition:

**Definition 2.** *Let $A = I(v)$ be a cluster based on a class $v$. An **outlier** for $A$ is any $o \in O_1(v) \cup O_2(v) \cup O_3(v)$. If $o$ is in $O_1(v)$, $O_2(v)$, or $O_3(v)$, we denote the **outlier class** of $o$ as $O_1$, $O_2$, or $O_3$ (respectively).*

Intuitively, the three outlier classes denote different degrees of 'dissimilarity' from the original cluster; $O_1$ outliers are the most challenging to distinguish, for they are semantically quite similar to the cluster. $O_2$ outliers are slightly easier to distinguish, and $O_3$ outliers should be quite simple to pick out.

The final dataset (a set of ⟨cluster, outliers⟩ pairs) is then created by serializing the following:

$$D = \tau \left( f_D \left( \bigcup_{c \in C} \langle f_i(\sigma_i[I(c)]), f_o \left( \sigma_o[O_1(c)] \cup \sigma_o[O_2(c)] \cup \sigma_o[O_3(c)] \right) \rangle \right) \right) \tag{16}$$

Where $\sigma_i$ and $\sigma_o$ are functions which select up to a given number of elements from the given set of instances and outliers (respectively), and $f_D$, $f_i$, and $f_o$ are functions which filter out dataset elements, instances, and outliers (respectively) based on any number of heuristics (see Section 3.1). Finally, $\tau$ takes the resulting tuples and resolves their QIDs to the appropriate surface strings.

The benefit of stating the dataset in the above terms is that it is highly configurable. In particular, different languages can be targeted by simply changing $\tau$ to resolve Wikidata entities to their labels in that language.

