# Peer review of "Automated Generation of Multilingual Clusters for the Evaluation of Distributed Representations"

_ICLR 2017 — rejected_

[Author Response · Philip Blair · 11 Nov 2016]
**Dataset Link**

It is mentioned in a footnote within the paper, but here is the link to the WikiSem500 dataset and a sample evaluation script:

[Official Review · AnonReviewer3 · rating 6 · confidence 3 · 12 Dec 2016]
**Data set seems interesting, but it's a pity that the automatic generation hasn't been scaled up.**

First, let me praise the authors for generating and releasing an NLP data set: a socially useful task.

The authors use an algorithm to generate a 500-cluster-per-language data set in semantic similarity. This brings up a few points.

1. If the point of using the algorithm is to be scalable, why release such a small data set? It's roughly the same order of magnitude as the data sets released in the SemEval tasks over the recent years. I would have expected something orders of magnitude larger.

2. The authors hand checked a small subset of the clusters: they found one where it was ambiguous, and should probably have been removed. Mechanical Turk can scale pretty well -- why not post-facto filter all of the clusters using MT? This is (in effect) how ImageNet was created, and it has millions of items.

3. Evaluating data set papers is an tricky issue. What makes a data set "good" or publishable? There are a number of medium-sized NLP data sets released every year (e.g., through SemEval). Those are designed to address tasks in NLP that people find interesting. I don't know of a data set that exactly addresses the task that the authors propose: the task is trying to address the idea of semantic similarity, which has had multiple data sets thrown at it since SemEval 2012. I wish that the paper had included comparisons to show that the particular task / data combination is better suited for analyzing semantic similarity than other existing data sets.

Two final notes:

A. This paper doesn't seem very well-suited to ICLR.  New NLP data sets may be indirectly useful for evaluating word embeddings (and hence representations). But, I didn't learn much from the paper: GloVe is empirically less good for semantic similarity than other embeddings? If true, why? That would be interesting.

B. The first proposal for the "put a word into a cluster and see if it stands out" task (in the context of human evaluation of topic models), is

Jonathan Chang, Jordan Boyd-Graber, Chong Wang, Sean Gerrish, and David M. Blei. Reading Tea Leaves: HowHumans Interpret Topic Models. Neural Information Processing Systems, 2009

which deserves a citation, I think.

[Official Review · AnonReviewer2 · rating 8 · confidence 4 · 14 Dec 2016]
**Accept**

This paper describes a new benchmark for word representations: spotting the “odd one out”. The authors build upon an idea recently presented at the RepEval workshop, but are able to collect a significantly larger amount of examples by relying on existing ontologies.

Although the innovation is relatively incremental, it is an important step in defining challenging benchmarks for general-purpose word representations. While humans are able to perform this task almost flawlessly (given adequate domain knowledge), the experiments in this paper show that current embeddings fall short. The technical contribution is thorough; the dataset construction appears logical, and the correlation analysis is convincing. I would like to see it accepted at ICLR.

[Official Review · AnonReviewer1 · rating 5 · confidence 3 · 16 Dec 2016]
**Borderline**

This paper introduces a new dataset to evaluate word representations. The task considered in the paper, called outlier detection (also known as word intrusion), is to identify which word does not belong to a set of semantically related words. The task was proposed by Camacho-Collados & Navigli (2016) as an evaluation of word representations. The main contribution of this paper is to introduce a new dataset for this task, covering 5 languages. The dataset was generated automatically from the Wikidata hierarchy.
Entities which are instances of the same category are considered as belonging to the same cluster, and outliers are sampled at various distances in the tree. Several heuristics are then proposed to exclude uninteresting clusters from the dataset.

Developing good ressources to evaluate word representations is an important task. The new dataset introduced in this paper might be an interesting addition to the existing ones (however, it is hard to say by only reviewing the paper). I am a bit concerned by the lack of discussion and comparison with existing approaches (besides word similarity datasets). In particular, I believe it would be interesting to discuss the advantages of this evaluation/dataset, compared to existing ones such as word analogies. The proposed evaluation also seems highly related to entity typing, which is not discussed in the paper.

Overall, I believe that introducing ressources for evaluating word representations is very important for the community. However, I am a bit ambivalent about this submission. I am not entirely convinced that the proposed dataset have clear advantages over existing ressources. It also seems that existing tasks, such as entity typing, already capture similar properties of word representations. Finally, it might be more relevant to submit this paper to LREC than to ICLR.

[Final Decision · Program Chairs · 06 Feb 2017]
**ICLR committee final decision**

The reviewers (and I) welcome new approaches to evaluation, and this work appears to be a sound contribution to that space. I suggest that, because the paper's incremental findings are of somewhat narrower interest, it is a good fit for a workshop, so that the ideas can be discussed at ICLR and further developed into a more mature publication.